Information from dynamic length changes improves reliability of static ultrasound fascicle length measurements

Aeles Jeroen jeroen.aeles@kuleuven.be 1
Lichtwark Glen A. 2
Lenchant Sietske 1
Vanlommel Liesbeth 1
Delabastita Tijs 1
Vanwanseele Benedicte 1
1 Department of Kinesiology, KU Leuven , Leuven , Belgium
2 School of Human Movement Studies, University of Queensland , Brisbane , Australia
Holmes Mike
Electronic publication date: 2017 Dec 15
Publication date: 2017
Volume: 5
Electronic Location ID: e4164
Received 2017 Sep 11; Accepted 2017 Nov 23
Copyright: ©2017 Aeles et al.
Copyright year: 2017
Copyright holder: Aeles et al.
License: This is an open access article distributed under the terms of the Creative Commons Attribution License, which permits unrestricted use, distribution, reproduction and adaptation in any medium and for any purpose provided that it is properly attributed. For attribution, the original author(s), title, publication source (PeerJ) and either DOI or URL of the article must be cited.
License URL: https://creativecommons.org/licenses/by/4.0/

Keywords: Inter-rater, Intra-rater, Repeatability, Muscle

Funding: The authors received no funding for this work.

==============================
Purpose

Various strategies for improving reliability of fascicle identification on ultrasound images are used in practice, yet these strategies are untested for effectiveness. Studies suggest that the largest part of differences between fascicle lengths on one image are attributed to the error on the initial image. In this study, we compared reliability results between different strategies.

Methods

Static single-image recordings and image sequence recordings during passive ankle rotations of the medial gastrocnemius were collected. Images were tracked by three different raters. We compared results from uninformed fascicle identification (UFI) and results with information from dynamic length changes, or data-informed tracking (DIT). A second test compared tracking of image sequences of either fascicle shortening (initial-long condition) or fascicle lengthening (initial-short condition).

Results

Intra-class correlations (ICC) were higher for the DIT compared to the UFI, yet yielded similar standard error of measurement (SEM) values. Between the initial-long and initial-short conditions, similar ICC values, coefficients of multiple determination, mean squared errors, offset-corrected mean squared errors and fascicle length change values were found for the DIT, yet with higher SEM values and greater absolute fascicle length differences between raters on the first image in the initial-long condition and on the final image in the initial-short condition.

Conclusions

DIT improves reliability of fascicle length measurements, without lower SEM values. Fascicle length on the initial image has no effect on subsequent tracking results. Fascicles on ultrasound images should be identified by a single rater and care should be taken when comparing absolute fascicle lengths between studies.

Introduction

It has long been established that muscle fascicle length changes are decoupled from the length changes of the entire muscle-tendon unit (Hoffer et al., 1989; Fukunaga et al., 2001). Because of this, inferences about fascicle behavior from kinematic data is difficult and can lead to errors, especially in muscle-tendon units with relatively long tendons. Objective data of muscle fascicle lengths is therefore needed. As such, an increased interest in identifying muscle fascicle geometry has been emerging, as muscle fascicle length and orientation can provide valuable information about muscle performance (e.g., Abe, Kumagai & Brechue, 2000; Blazevich, 2006; Farris et al., 2016a; Farris et al., 2016b) as well as clinical or training adaptations (Mohagheghi et al., 2007; Blazevich et al., 2014; Hoffman et al., 2016). B-mode ultrasound imaging is the most commonly used method to determine muscle fascicle geometry, because it is cost- and time-effective and it allows for measurements during dynamic tasks, such as walking (Fukunaga et al., 2001; Ishikawa et al., 2005), running (Lichtwark & Wilson, 2006; Ishikawa, Pakaslahti & Komi, 2007), and jumping (Kurokawa, Fukunaga & Fukashiro, 2001; Farris et al., 2016a; Farris et al., 2016b).

Acquired ultrasound images are often analyzed by manually identifying muscle fascicles to obtain information about fascicle length and orientation (Cronin et al., 2011; Cronin & Lichtwark, 2013). This has the potential to result in low reliability and low repeatability because of the subjective nature of this analysis. Studies on the reliability of fascicle identification from ultrasound images during a wide range of tasks, including measurements of muscle in a relaxed and contracted state, during walking, running and jumping, have reported standard error of measurement (SEM) percentages of 4.3–14.2% for inter-session (Kwah et al., 2013), 0.0–8.3% for inter-image (Kwah et al., 2013) and 3.8–7.5% for inter-rater (König et al., 2014; McMahon, Turner & Comfort, 2016) analyses. Overall, these values remain rather high, considering the effect sizes generally reported in cross-sectional or longitudinal training studies (10–19%) (Abe, Kumagai & Brechue, 2000; Fukutani & Kurihara, 2015; Timmins et al., 2016) and in studies that compare fascicle length changes between various conditions within similar dynamic tasks (9–14%) (Lichtwark & Wilson, 2006; Farris & Sawicki, 2012; Brennan et al., 2017). It is therefore essential to explore methods for lowering fascicle identification errors and increasing reliability.

Even though manual fascicle identification remains the “gold standard”, it is likely to induce subjective errors, for example due to experimenter bias (Cronin et al., 2011). Various efforts have been made in the last two decades to objectify fascicle tracking on image sequences by development of (semi-) automated image-processing algorithms. These algorithms are mainly based on cross-correlation methods (Loram, Maganaris & Lakie, 2004; Herbert et al., 2011) or optical flow methods (Magnusson et al., 2003; Rana, Hamarneh & Wakeling, 2009; Cronin et al., 2011; Farris & Lichtwark, 2016) and allow for an automated tracking of visible structures on ultrasound image sequences. Most of these proposed automated methods have proven to accurately match manual trackings and allow for a more objective analysis. This makes automated processing an appealing alternative for manual processing as it is also less time-consuming and thus more efficient. In spite of these advantages of automated tracking, the errors made between consecutive trackings still remain rather high. Gillet, Barrett & Lichtwark (2013) reported standard errors between 5% and 10% of absolute fascicle length, similar to manual tracking (Gillet, Barrett & Lichtwark, 2013; Kwah et al., 2013). They stated that this is most likely contributed to by the errors made on the required initial manual input, which is often required on the first frame of an image sequence for this type of automated tracking algorithm (Herbert et al., 2011; Cronin et al., 2011; Gillet, Barrett & Lichtwark, 2013; Farris & Lichtwark, 2016). These authors suggest that the initial length estimate variability likely explains the greatest part of variability in fascicle tracking reliability and recommend efforts for improvement of the initial fascicle tracking.

Many studies on static muscle fascicle architecture use single-image ultrasound recordings (Abe, Kumagai & Brechue, 2000; Karamanidis et al., 2011; Franchi et al., 2014; Aeles et al., 2017). For these studies it may be of even greater importance to find methods for improving the fascicle identification reliability, as the absolute fascicle lengths are generally the main outcome. A natural consequence of this single-image method is that there is no information prior to or after the initial image to help guide the researcher for accurate and reliable fascicle identification. It is often believed by researchers in this particular field that tracking of image sequences is more reliable than fascicle identification on single images as the researcher can focus on movement patterns to identify the correct fascicle orientation on the image sequences. Despite the recommendations by Gillet, Barrett & Lichtwark (2013), no studies, to the best of our knowledge, have focused on improving fascicle identification reliability for single-image and image sequence ultrasound recordings.

The aim of this study was therefore, to compare different strategies for ultrasound fascicle length measurements, commonly used in practice, in order to increase fascicle identification reliability between different raters. We used images of the medial gastrocnemius as this is a popular muscle for ultrasound measurements in vivo because it is a superficial muscle with relatively short muscle fascicles and it has important contributions during tasks such as walking, running and jumping (Fukunaga et al., 1997; Lichtwark & Wilson, 2007). Our first strategy test compared images from single-image fascicle identification, with no information before and after the image, i.e., uninformed fascicle identification (UFI) and the first images of image sequences with fascicle information after the initial image, i.e., data-informed tracking (DIT). We hypothesized the UFI to yield larger between-rater differences and lower reliability compared to the DIT. For our second strategy test, we compared ultrasound image sequences with two different fascicle starting lengths. We hypothesized lower between-rater differences and greater reliability when tracking fascicle shortening (initial frame has long fascicle lengths; further named ‘initial-long condition’) compared to tracking fascicle lengthening (initial frame has short fascicle lengths; further named ‘initial-short condition’). This hypothesis was based on the premise that fascicles are more clearly visible at long lengths, when they have their lowest pennation angle and therefore the fascicle structure is more perpendicular to the ultrasound waves, creating more defined images (Lichtwark, 2017).

Materials & Methods

Participants

Ultrasound image sequences of passive ankle joint rotations were collected for 28 participants (13 female, 15 male; body height = 179.73 ± 8.21 cm; body mass = 73.01 ± 9.21 kg). From these 28 participants, five were randomly selected and single-image ultrasound recordings were taken for an intra-rater reliability test and the UFI analyses. For the DIT comparisons, all 28 participants were included. An overview of data collection is shown in Fig. 1. All participants confirmed to participate in the study by written informed consent. Three independent researchers, who will be referred to as ‘raters’ were asked to participate in the study for the fascicle identification of the ultrasound images. All raters were experienced with identification and tracking of medial gastrocnemius fascicles and were not informed about the purpose of the study. The study was approved by the local ethics committee (ethische commissie onderzoek UZ / KU Leuven; approval number - S57477 - ML11371) and conforms to the recommendations of the Declaration of Helsinki.

Figure 1 Protocol overview.

Graphical summary of the data collection and further categorisation of the different datasets for final analyses. DIT, data-informed tracking; UFI, uninformed fascicle identification. B shows the comparison between initial-long and initial-short conditions for the DIT condition. A shows the DIT and UFI comparison.

Experimental protocol

For the assessment of the intra-rater reliability and the UFI analysis, single-image recordings of the medial gastrocnemius fascicles of the left and right leg were collected for five subjects. For each leg and subject, three images were taken at approximately the same mid-image location of the muscle with the ankle joint in a maximal dorsiflexed position. The fascicle length at this joint angle is equal to the fascicle length used in the initial-long condition for the DIT and thus these images were used to compare the UFI and DIT strategies. For the DIT and image sequence tracking analyses, ultrasound images of the medial gastrocnemius fascicles of the left and right leg were collected during passive rotations of the ankle joint. Subjects were laying in prone position on a table, with the knee and hip joint fully extended. During this passive trial, the ankle joint was manually rotated three times over the full range of motion with the subject fully relaxed. All ultrasound image recordings were captured using a Telemed Echoblaster 128 CEXT system (UAB Telemed, Vilnius, Lithuania). B-mode images were collected at 30 Hz for all measurements using a 60 mm, 128-element linear transducer (LV 7.5/60/128Z-2; UAB Telemed, Vilnius, Lithuania). The transducer was positioned longitudinally over the mid-belly of the medial gastrocnemius for all measurements. During imaging, the transducer was held at the same location by the operator. Minimal movement of the transducer during the ankle joint rotations was confirmed with markers on the transducer and on the knee joint using a motion capture system (Vicon, Oxford Metrics, UK).

Image processing protocol

All ultrasound images were processed using fascicle identification software (Farris & Lichtwark, 2016) in MATLAB R2014 (The Mathworks, Natick, MA, USA). For this study, all raters were asked to identify the fascicle end-points manually on each frame. All raters used the same fascicle identification techniques: at first, two lines were manually drawn on the image, one over the deep aponeurosis and one parallel to the muscle fascicles with attachments to the deep and superficial aponeurosis. The length of this line represented the fascicle length and was calculated based on its relative length to the image depth, which was set during the measurements at 50 mm. Each rater received the same instructions for the fascicle identification: they were asked to identify the initial image of each image sequence first and then, for the image sequences, manually track the fascicle by adjusting the fascicle end-points on each frame. For the image sequences, the raters were allowed to watch the sequences prior to fascicle identification and to play the sequences back and forth during the tracking. Raters could adjust the fascicle end-points on the initial image and on all subsequent images based on the fascicle information they obtained from the dynamic length changes of the fascicle on the subsequent frames (i.e., DIT). For all images, raters were instructed to focus on the middle region of the image for identification of the fascicle, as fascicle behavior may differ throughout the muscle belly (Lichtwark & Wilson, 2007) and this ensures that most of the fascicle is visible in the image. For the fascicles with attachments outside of the image, visual linear extrapolation of the superficial aponeurosis only was used by the raters, ensuring that at least the attachment on the deep aponeurosis was visible throughout the whole image sequence. For the tracking of the passive rotation image sequences, the second of three full rotation cycles was extracted and split at the maximum fascicle length, resulting in one image sequence file with fascicle shortening (=initial-long) and one with fascicle elongation (=initial-short).

All raters followed the same order of analyzing the different ultrasound recordings. They were instructed to first track the static ultrasound images used for the reliability test and UFI analysis. Afterwards, each rater tracked the image sequences of the passive rotations. All raters first tracked all files containing the initial-long images and then all files containing the initial-short. Due to technical issues (e.g., no data of full ankle joint range of motion or missing data from one of the raters), three files from the initial-long condition and 6 files from the initial-short condition were excluded. As such, 53 sets of initial-long and 50 sets of initial-short image sequences were used for further analyses. In order to exclude tracking biases, all data was randomized per set of recordings and blinded for the raters. After data processing, fascicle length results from the left and right leg were combined for all data sets for analyses.

Data analyses

For the comparison between UFI and DIT, inter-rater intra-class correlations (ICC) (2,1; single) and SEM values were calculated between the different raters using SPSS v.22 software (IBM SPSS, Armonk, NY, USA). This was done using all of the 30 single-image recordings for the UFI condition and the first image of each of the 53 initial-long image sequences for the DIT condition.

Each of the analyses further described were done for both initial-long and initial-short fascicle conditions. All image sequence waveforms were low-pass filtered using a fourth order Butterworth filter (MATLAB R2014; The Mathworks, Natick, MA, USA). The initial-long and initial-short strategies were compared for the DIT condition, using the first image of each of the 53 initial-long and the first image of each of the 50 initial-short image sequences. Coefficients of multiple determination (CMD) were calculated between the waveforms of two of the raters and the corresponding waveform of a reference rater. The reference rater was chosen based on the results of an intra-rater ICC (2,1; single) and SEM analysis. For this, 10 of the 30 single-image recordings that were used for the UFI, were analysed three times non-consecutively by each rater for the assessment of the intra-rater reliability. ICC values for all raters were good to excellent with the lowest ICC equal to 0.78. SEM values were good to excellent with a maximal value of 3.73 mm. The ICC value of the reference rater was very high (0.98) and SEM was low (0.81 mm). Calculating the CMD values between the reference rater and each of the other raters resulted in two CMD values per set, which were first averaged per set and then averaged over all sets. To assess the absolute error, the mean squared error (MSE) was calculated. MSE was calculated as the mean of the squared difference between the waveforms of each of the rater at every data point. Again, MSE values were first averaged per set and then averaged over all sets. To test for the influence of the variability between raters on the initial image, we used the methods described by Gillet, Barrett & Lichtwark (2013): all waveforms were corrected for their respective initial fascicle length and MSE analyses was repeated (=offset-corrected condition) (Fig. 2B). This was done by subtracting the fascicle length of the first image from each data point of the respective waveforms.

Figure 2 Analyses example.

Example set of three waveforms without (A) and with (B) the initial-frame offset removed, one from each rater, for the initial-long condition. The black solid waveform is the tracking from the reference rater. Data analyses methods and results are shown for (the differences between) the grey dashed waveform and the black solid waveform as an example. Note that the values in the results section are average differences between all three waveforms.

To evaluate the practical relevance and impact of the different strategies, the differences between raters for physiologically-relevant parameters were calculated (Fig. 2A). A first parameter was the total fascicle length change over the full ankle joint range of motion, which was calculated as the difference between the longest and shortest length of the fascicle on one waveform. Secondly, to test for outcome differences in static muscle architecture studies that use single-image recordings, absolute fascicle lengths were compared for the three raters on the first and last image of each waveform. Results were calculated both as absolute and relative differences between the two respective raters. The relative differences were calculated as the percentage of the average absolute fascicle length between the two respective raters. For all these parameters, the difference between the three raters were calculated per set of waveforms, after which these differences were first averaged per set of waveforms and then over all sets.

Results

ICC values were greater for the DIT as compared to the UFI, however similar SEM values were found (Table 1). Between the initial-long and initial-short conditions, similar ICC values were found for the DIT. However, a higher SEM value was found in the initial-long condition compared to the initial-short condition.

Table 1 DIT–UFI comparison results.

Intra-class correlations and standard error of measurement values (mm) for the data-informed tracking (DIT) and uninformed fascicle identification (UFI).

Condition		DIT	UFI	
Initial-long	ICC	0.818	0.666	
SEM	4.15	4.29	
Initial-short	ICC	0.857	/	
SEM	3.24	/	

CMD values were very high for both the initial-long and initial-short conditions and not significantly different between both conditions (Table 2). MSE values were not significantly different between the two conditions for both the original waveforms as well as the offset-corrected waveforms. However, MSE significantly decreased after offset correction (p < 0.001) in both conditions.

Table 2 CMD results for the initial-long and initial-short conditions.

Values are means ± SD. Coefficient of multiple determination (CMD) values approaching 1 denote high similarity of waveforms. Mean squared error (MSE) values are in mm.

Condition	R2	MSE	MSE [offset-corrected]	
Initial-long	0.98 ± 0.05	5.07 ± 2.61*	2.75 ± 1.38	
Initial-short	0.98 ± 0.02	5.30 ± 2.36*	2.52 ± 1.52	
Notes.

* Significantly different compared to the offset-corrected condition (p < 0.05).

Differences in total fascicle length change between the reference rater and raters 1 and 2 are shown in Fig. 3. Neither absolute nor relative differences between raters were significantly different between the initial-long and initial-short conditions (Tables 3 and 4). Absolute differences in fascicle length between the different raters were greater on the initial image in the initial-long condition compared to the initial-short condition (p = 0.005). This difference was non-existent when comparing the relative differences. On the final image, the absolute differences in fascicle length were smaller in the initial-long condition compared to the initial-short condition (p = 0.003). Again, this was not found when comparing the relative differences.

Figure 3 Bland-Altman plot.

Bland-Altman plot of the difference in total fascicle length change during the passive rotation between rater 1 (R1) and the reference rater (Rref) (black *) and rater 2 (R2) and Rref (grey ♢) on the y-axis. X-axis shows the total length change values of Rref for the initial-long condition. Mean differences between R1 and Rref are shown by the black solid line and between R2 and Rref by the grey solid line. The black dashed lines give the upper and lower boundary of 1.96 times the standard deviation (SD) for the difference between R1 and Rref, the grey dotted lines show this for the difference between R2 and Rref.

Table 3 Absolute differences between raters for physiologically-relevant parameters for the initial-long and initial-short conditions.

Values are means ± SD. All values represent the absolute differences between values of the different raters for that respective parameter. Fascicle length differences are in mm.

Condition	Total fascicle length change	Fascicle length - first image	Fascicle length - final image	
Initial-long	4.83 ± 2.56	6.14 ± 3.16a	4.42 ± 2.58a	
Initial-short	3.97 ± 2.80	4.54 ± 2.43	6.15 ± 3.11	
Notes.

a Significantly different from the initial-short condition.

Table 4 Relative differences between raters for physiologically-relevant parameters for the initial-long and initial-short conditions.

Values are means ± SD. All values represent the relative differences between values of the different raters for that respective parameter in % of the absolute total fascicle length change and absolute fascicle length on the first and final image respectively.

Condition	Total fascicle length change	Fascicle length - first image	Fascicle length - final image	
Initial-long	34.29 ± 17.96	8.98 ± 5.12	8.34 ± 4.97	
Initial-short	31.01 ± 18.86	8.12 ± 4.29	8.89 ± 4.55	

Discussion

In this study, we tested different strategies for ultrasound fascicle length measurements that are commonly being used in practice. Our first aim was to compare two strategies for use in studies that make use of single-image recordings for fascicle length. We made a comparison between single-image recordings with no information before or after the image of interest (UFI) and the initial frame of image sequences with information after the image of interest (DIT). There was a substantial difference between the DIT and UFI in ICC scores. However, SEM values were not different between the two conditions, suggesting that although the identification error was not influenced by the condition, the error was made in a more consistent manner. Raters often rely on the movement of the fascicle and changes in its orientation and length to help identify the correct movement patterns during fascicle identification. As this cannot be done on single-image fascicle lengths, our results suggest that it is worthwhile to record image sequences with fascicle movement, even for studies interested in single-image analyses. SEM values between different raters remain rather high (4.15 and 4.29 mm for DIT and UFI respectively) for both strategies, especially compared to intra-rater SEM values (average of 2.59 ± 1.56 mm), but are within the range reported in other studies (König et al., 2014; McMahon, Turner & Comfort, 2016).

As the DIT strategy proved to be a more reliable method for single-image outcomes and it can be used in studies interested in fascicle length changes during dynamic activities, we aimed at further improving this method by employing our second strategy. For this strategy we compared two conditions with different fascicle lengths on the initial image, either initial-long or initial-short. ICC values for both conditions were similar, yet against expectations, the SEM value for the initial-long condition was almost 1 mm greater compared to the initial-short condition. A reasonable explanation for this is that many times, in the initial-long images, fascicle attachments to the superficial aponeurosis are outside of the visible image area. In these cases, linear extrapolation of both the tracked fascicle and the aponeurosis was used visually by the raters, a common method in fascicle length measurements for ultrasound images. As such, efforts should be made to avoid this type of error, for example by using suggested methods in which only the visible part of the fascicle is identified and then the whole fascicle length is calculated using extrapolation in the analyses only after identification of the fascicle on the image (Finni et al., 2003; Seiberl et al., 2010). However, even though this method is very useful in muscles with minimal fascicle curvature such as the medial gastrocnemius, they should be used with care in other muscles such as the biceps femoris, which shows fascicles with substantial curvature (as shown on Fig. 1A in Seymore et al., 2017). To verify that this greater difference in fascicle length between the raters was actually due to the longer fascicle length together with the associated difficulties that were addressed earlier, and not due to the fact that this was the initial frame of the image, we compared the absolute MSE values between the raters on both the initial and final image of each waveform. Here we again found a significant difference between the two conditions, with a larger difference between the raters in the initial-long compared to the initial-short condition on the first image but the opposite, a larger difference between the raters in the initial-short compared to the initial-long conditions, on the final image. As such, it appears that this difference is indeed mainly a result of the difference in absolute fascicle length at that respective image, as also suggested by Gillet, Barrett & Lichtwark (2013). Indeed, our results clearly show that the greater difference between raters on images with long fascicle length is due to the greater fascicle length and is independent of its relative position (i.e., first or final image) in the waveform, as the differences relative to the average fascicle length on that image were equal between the initial-long and initial-short condition for either the first (8.98 ± 5.12% versus 8.12 ± 4.29% respectively) or last (8.34 ± 4.97% versus 8.89 ± 4.55% respectively) frame.

Since many movements commonly assessed with ultrasound are cyclical movements, such as gait or passive and active ankle joint rotations, researchers are often allowed to choose at which part of an image sequence cycle they would like to start their fascicle tracking. Through personal communication, we established that many researchers in practice prefer to start tracking on either shortened fascicles, preferring the initial-short strategy or lengthened fascicles, preferring the initial-long strategy as it is commonly believed that this results in lower fascicle identification errors for the rest of the image sequence. The longer fascicle in the initial-long strategy in pennate muscles is generally accompanied by a lower pennation angle, decreasing the angle of incidence of the sound waves sent out by the transducer and allowing more reflections of the sound waves. This generally increases the visibility of the imaged structures (Lichtwark, 2017). However, our results indicate no difference in inter-rater reliability, inter-rater waveform similarity (CMD), absolute differences between raters (MSE), and total fascicle length change differences between the two conditions. Overall, these results suggest that the length of the fascicle on the initial image does not influence the reliability of the subsequent tracking in image sequences. However, it must be noted that the image sequences used in this study were not cyclical, as only one joint rotation was used for analysis and cut at the maximum fascicle length to obtain the initial-short and initial-long image sequences. Image sequences of cyclical motions that have multiple waves introduce the advantage of the fascicle returning to a similar length, allowing the rater to compare the fascicle lengths at similar sections of the wave and adjusting the fascicle identification accordingly.

The conclusions in this study were drawn from our analyses of the medial gastrocnemius muscle fascicles. However, other muscle-tendon units often present different fascicle geometries, for example the knee extensors have much longer fascicles and more curvature of the fascicles (Seymore et al., 2017). As the strategies discussed in this paper have not been tested in these muscle-tendon units, we should be careful in generalizing these results. For example, due to the longer fascicles in the vastus lateralis, it is likely that large portions of the fascicle lies outside the image when using short transducers (e.g., the 60 mm transducer used in this study), especially when stretching the fascicles to long lengths. As such, in these muscle-tendon units the effect between the initial-short and initial-long condition may be more significant than shown by the results in the current study.

When we corrected the fascicle length on each frame for the fascicle length on the initial image, MSE was significantly lowered. This shows that a large part of the variability between fascicle lengths of different raters is explained by the variability in fascicle length on the initial image. This finding was also reported by Gillet, Barrett & Lichtwark (2013) when comparing automated trackings with different initial-image inputs by different raters. Combined with the very high CMD values found in this study, we can conclude that there was high similarity between the tracking of waveforms by different raters and that the main difference arises in the initial fascicle length estimation. As such, relative length changes of the fascicle are highly reliable between different raters, but less so in terms of absolute values.

Even though offset-correction is a successful strategy for lowering MSE values between different raters, it has no effect on physiologically-relevant parameters such as the total length change of the fascicle. To our knowledge, no other study has tested the reliability between raters for analyses of total fascicle length changes on ultrasound images. Yet the observed differences in these length changes between raters in this study are in close proximity to reported effect sizes in total fascicle length changes (4–6 mm) of other studies (Duclay et al., 2009; Sakuma et al., 2012; Theis et al., 2013; Blazevich et al., 2014). Indeed, as can be seen on Fig. 3, these effect sizes are well within the range of the coefficients of repeatability (calculated as 1.96 times the standard deviation of differences between two raters). This means that there is a 95% probability that differences in total fascicle length change between raters are greater than the actual effect size. As such, we suggest that processing of image sequences in studies that are interested in parameters such as absolute fascicle length changes, should not be performed by different raters, as could sometimes be preferable in large studies. Furthermore, this means that care should be taken when comparing absolute values of fascicle length changes between different studies. However, as we did not test within-rater reliability on the image sequences, we cannot conclude that processing of the images by a single rater is more reliable. Yet it seems that the differences between the raters are rather consistent (Fig. 3) in terms of over- or underestimating the total fascicle length change compared to the other raters. Together with the high CMD values between the waveforms of each set, this suggests that raters are consistent in their tracking both within one image sequence as well as between the different image sequences.

In this study we have only performed analyses on static or passive ultrasound images. During active movements, the fascicle pennation angle can be higher compared to our initial-short condition, potentially causing reduced image quality. Yet as we did not find any differences between the initial-short and initial-long condition, we speculate that this will be true for active conditions, especially at similar pennation angles compared to maximal passive plantarflexion. Furthermore, we assume that dynamic contractions can be used for the DIT method instead of using passive movements. One limitation of the current study is the low number of subjects used for the UFI condition. Inclusion of extra subjects would have increased the statistical power for these analyses, yet by using both legs, power was slightly increased, as both legs can be considered as independent measures due to the large bilateral differences in muscle fascicle architecture (Aeles et al., 2017).

Conclusions and recommendations

In conclusion, we have shown that DIT does not result in lower SEM values but a higher reliability between the raters was found, suggesting a more consistent fascicle identification. SEM values were lower however when using initial-short image sequences, yet this effect may be cancelled when calculating differences relatively to the absolute fascicle length. Overall, studies interested in single-image fascicle lengths are advised to record image sequences with fascicle length changes prior to or after the fascicle length of interest. For these image sequences, we have shown that the difference in fascicle length on the initial image explains most of the variability between the fascicle lengths of different raters. Differences between raters in fascicle lengths of image sequences are still relatively high, yet when calculated relative to the fascicle length they appear independent of the fascicle length on the initial image and the results show high similarity and consistency. However, caution is needed when comparing fascicle lengths between different studies, mainly in terms of absolute values (e.g., total fascicle length changes). Studies that look at fascicle length changes over time, such as longitudinal training studies, should take these findings into account and should look to achieve high consistency between pre and post measurements and maximize the reliability between pre and post analyses. For example, Aeles et al. (2017) suggested that muscle fascicle lengths should be measured at a muscle-tendon unit length where there is no tension on the muscle-tendon unit and showed large variations between subjects in the ankle joint angle at which this is true for the medial gastrocnemius. It remains unknown whether this joint angle changes following a training intervention. As such, these joint angles should be determined for each individual in both the pre and post measurement. Additionally, by applying the DIT method and having the fascicle identification done by a single rater, the reliability of the analysis can be increased. Furthermore, both the intra-rater error as well as the inter-rater error should be taken into account when drawing conclusions from comparisons of absolute fascicle lengths in both static and dynamic conditions either within one study or between different studies. SEM from our raters ranged from 1.5 to 5% of absolute fascicle length within one rater and was around 7% between different raters. These values are not far from the typically reported effect sizes mentioned in the introduction and are within the range of previously reported SEM values. Our analyses on relative differences between raters in absolute fascicle length showed average values of 8.12–8.98%. As such, we recommend caution when drawing conclusions from fascicle length comparisons with differences below these values of roughly 9% of absolute fascicle length. Combined, our findings suggest that fascicle identification on ultrasound images is best done by one rater and that the error due to the manual fascicle identification should be taken into account when comparing absolute fascicle lengths between different raters or between different studies. Finally, we urge that other strategies for improvement of the initial image fascicle identification on ultrasound images should be studied.

Supplemental Information

Supplemental Information 1 Raw data

Matlab file with raw data. This file contains 4 variable structures (’Data’, ’Results_Final’, ’Stats_Data’, and ’Stats_Results’). These structures further contain several substructures containing several variables. All variables should be clear and comparable with the parameters/results in the manuscript. ’Data’ contains the raw data. ’Results_Final’ contains the results after analyses. ’Stats_Data’ contains the data as prepared for the statistical analyses. ’Stats_Results’ finally contains the p-values of the statistical analyses.

Click here for additional data file.

Supplemental Information 2 Data example, Rater 1

Example video for the fascicle tracking by the reference processor for an ‘initial-long’ ultrasound image sequence. Note that this video contains more image frames than were used for analyses. Data for analyses was extracted somewhere between the frames at which the ‘fascicle tracking line’ is drawn horizontally (somewhere between 6 and 8 seconds of total video length).

Click here for additional data file.

Supplemental Information 3 Data example, Rater 2

Example video for the fascicle tracking by the third (non-reference) processor for an ‘initial-long’ ultrasound image sequence. Note that this video contains more image frames than were used for analyses. Data for analyses was extracted somewhere between the frames at which the ‘fascicle tracking line’ is drawn horizontally (somewhere between 6 and 8 seconds of total video length).

Click here for additional data file.

Supplemental Information 4 Data example, Rater 3

Example video for the fascicle tracking by one of the (non-reference) processors for an ‘initial-long’ ultrasound image sequence. Note that this video contains more image frames than were used for analyses. Data for analyses was extracted somewhere between the frames at which the ‘fascicle tracking line’ is drawn horizontally (somewhere between 6 and 8 seconds of total video length).

Click here for additional data file.

Abbreviations

CMD Coefficients of multiple determination

DIT Data-informed tracking

ICC Intra-class correlations

MSE Mean squared error

SEM Standard errors of measurement

UFI Uninformed fascicle identification

Additional Information and Declarations

Competing Interests

Author Contributions

Human Ethics

Data Availability

The authors declare there are no competing interests.

Jeroen Aeles conceived and designed the experiments, performed the experiments, analyzed the data, wrote the paper, prepared figures and/or tables, reviewed drafts of the paper.

Glen A. Lichtwark conceived and designed the experiments, reviewed drafts of the paper.

Sietske Lenchant and Liesbeth Vanlommel performed the experiments, analyzed the data.

Tijs Delabastita analyzed the data, reviewed drafts of the paper.

Benedicte Vanwanseele conceived and designed the experiments, contributed reagents/materials/analysis tools, reviewed drafts of the paper.

The following information was supplied relating to ethical approvals (i.e., approving body and any reference numbers):

The study was approved by the local ethics committee (ethische commissie onderzoek UZ / KU Leuven; approval number - S57477 - ML11371) and conforms to the recommendations of the Declaration of Helsinki.

The following information was supplied regarding data availability:

The raw data is provided in the Supplemental Files.

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
