# Peer review of "Information from dynamic length changes improves reliability of static ultrasound fascicle length measurements"

_PeerJ, doi:10.7717/peerj.4164_

## Round 0.1 · original submission · Minor Revisions

· Academic Editor

Minor Revisions

Dear Authors,

Your manuscript review has now been completed for the manuscript "Information from dynamic length changes improves reliability of static ultrasound fascicle length measurements". You will notice that we had three excellent and comprehensive reviews of your work and I believe that these comments can help improve the quality of this manuscript.

While the reviewers believed that the manuscript was interesting and the study relevant, you will find some commonalities between reviewers. Some concerns around methodology and analysis techniques are presented that need to be addressed prior to being considered for publication in PeerJ. Reviewer comments can be found below. We look forward to your reply.

Regards,

Reviewer 1 ·

Basic reporting

In general, the description of the protocol is difficult to follow. For example, how can five subjects times two legs time three raters equal 56 sets of images (L200)? Please consider including a sketch summarising the protocol or use a more streamlined description.

Experimental design

See general comments

Validity of the findings

See general comments

Comments for the author

The present article compared different strategies to digitise muscle fascicle length from ultrasound images. The main findings are that informed digitisation (via analysis of preceding or consecutive frames) yielded more reliable results. The results also show that estimating changes in fascicle length in a series of images is more reliable than the assessment of single images when considering inter-rater differences.
The manuscript is globally well written and the study is interesting to the increasing number of scientists measuring fascicle length from ultrasound images. The conclusions are consistent with motor control theories highlighting the importance of optic flow in visuomotor tasks. My main question would be regarding the execution of the analysis and, by extension, the possibility of a missing experimental condition: In the sequence conditions, were the raters instructed to refrain from displaying the preceding images, playing the sequence back and forth? This strategy would be different than the strict, direct, digitisation of each image without playback, providing the rater with an optic flow, instead of discrete cues. In relation to this, it would have been interesting to propose another DIT condition, in which the frame of interest would have been in the middle of a sequence.

Minor comments:

Why taking 5 subjects out of 28?

The increased fascicle tension induced by maximal dorsiflexion considerably improves the echo and contrast from these structures. This aspect should be included in the factors making fascicle segmentation easier in these images (i.e. it is not the sole effect of fascicle orientation).

·

Basic reporting

The language is clear throughout the whole manuscript. References provided are appropriate and sufficient. The state of the figure is more than acceptable (although I will suggest to insert a new figure to better explain the methodology used).

Experimental design

The experimental design is clear but sometimes it requires a lot of focus by the reader to actually visualise how the study was conducted. I suggest the authors to provide a schematic representation of the scan acquisition between the two groups.
The rest of the method section is good.
Please specify the probe length and how the operator were sure that the probe was place on fascicle plane (I'm sure it was, just better specify in the methods you actually took care of this, especially during the passive movement).

Validity of the findings

The novelty of findings is appreciated, especially for a measure that we know being subjected to a lot of variability between many operators and also investigations. I have just a couple of points of discussion.

1- This was conducted on GM muscle, a muscle-tendon unit (MTU) that present a long and compliant tendon with a relatively short fascicle if compared to other muscle groups. For example, configuration of Quads (VL in the specific ) MTU may be different, as curvature of fascicle could differ along the length of the whole muscle, especially close to the myotendineous junction. While I appreciate and not question the validity data presented on GM, what would the authors think it could happen in a MTU with different configuration? This should be addressed in the discussion

2- What do the authors think of the influence of exercise adaptations in a scenario with pre-to-post changes in fascicle length? There is evidence for contraction dependent adaptations are found for example between eccentric vs. concentric loading type (done with single snapshot ultrasound without the use of DIT methodology as this paper advocates) (see Franchi 2014 and 2017, and I hope, then, that those data (mine) are actually saying the truth, then! ha ha) and we know that the length of fascicle can increase also because of greater fascicle curvature that can be found close to the superficial aponeourosis . Can this new methodology implement also to better identify where the real end of the fascicle presenting curvature is on the superficial aponeurosis? Maybe this could be a suggestion for a comment in the discussion or an actual study. And if so, you know who I am and I like to see if something can be done on VL too!

3- practical applications is then to see dynamic behaviour of muscle fascicle first. Can this be obtained with just a flash isometric contraction too, without moving the muscle throughout the whole range of motion?

Comments for the author

I enjoyed this paper and the advance in the methodology that this can bring.
I thank the authors for a nice read and an entertaining explanation of a new methodology advice that could actually make fascicle length assessment more reliable.

Reviewer 3 ·

Basic reporting

The authors provide a clear and unambiguous manuscript with sufficient literature and background context. Ultrasound imaging of skeletal muscle is an emerging technique to understand dynamic muscle behaviour and is of general interest to the biomechanics, muscle mechanics fields. The article structure is sound and the hypotheses are well laid out.

Experimental design

The experimental design is adequate and the research question is well defined. The techniques and ethical standards are indeed met. There are some additional details of the methods utilized that I have suggested in the general comments section which the authors should clarify in the manuscript.

Validity of the findings

The data is statistically sound, however I would like to see some additional analysis performed (as outlined in the general comments). I have some suggestions regarding the conclusions for the authors which, in my opinion, would strengthen the manuscript.

Comments for the author

Information from dynamic length changes improves reliability of static ultrasound fascicle length measurements

SUMMARY OF WORK:
In this study, the authors examine different strategies for manually tracking medial gastrocnemius fascicle length from B-mode ultrasound images during passive ankle rotations. Specifically, they compare fascicle lengths determined from a single static image, termed uninformed fascicle identification (UFI), to fascicle lengths determined from tracking a series of images allowing the identifier to adjust the initial fascicle length, termed data-informed tracking (DIT). They compared results from 3 different raters as well as from conditions where the fascicle was at an initial short length versus an initial long length. Their results show that DIT improves the reliability of fascicle length measurements and suggest caution when measuring fascicle lengths from one single ultrasound image as well as when comparing absolute fascicle lengths across different studies. The manuscript is well-written and would be of interest to ultrasound researchers. I suggest a number of things below that the authors should address in order improve the clarity/accuracy/completeness of the manuscript.
* * *
COMMENTS:

ABSTRACT:

1. Line 40: It is unclear what the authors are referring to by “four strategies.” Do you mean UFI, DIT, initial long and initial short? I am under the impression that UFI was compared to DIT only for a single frame (ie: the initial frame) whereas DIT initial long was compared to DIT initial short for a passive cycle (many frames). This sentence needs clarifying as the four strategies are not for the same condition.

INTRODUCTION:

2. Line 65: I’m not sure that ‘seriously erroneous’ is the correct word choice. Inferences about fascicle behaviour from kinematics alone is challenging in MTUs that contain considerable SEE, but may provide information about fascicle behaviour in MTUs with negligible external tendon. I suggest rephrasing.

3. Please clarify in the first paragraph of the introduction that you are referring to B-mode ultrasound.

4. Line 72: I suggest replacing “in dynamic settings” with “during dynamic tasks”

5. Line 96: I suggest using “optical flow algorithms” rather than “…modelling”
6. Line 117: Please clarify to the reader what you mean by “in practice”

7. Line 136-126: Please include why the medial gastrocnemius is popular for ultrasound studies.

METHODS:

8. Line 145: why was 5 out of the total 28 participants chosen for the analysis? Indeed a larger sample would improve the statistical power. This requires some clarifying.

9. Line 167: how was the ultrasound probe secured to the participant during the experiments?

10. Line 170-174: To my knowledge the ultrasound images were not “processed using a semi-automated algorithm” as is stated in the manuscript. This is partly misleading. I am under the impression that the authors used the Ultratrak GUI to manually track all images (both single frame and image series). Was Ultratrak in fact used to initially track the image sequences and then the observer then clicked through tracking results for each frame to update the locations of the upper and lower fascicle insertion sites? If so, this needs to be clarified. If not, it is unclear why Ultratrak was used rather than ImageJ or another image processing software. Please clarify.

11. Line 180-184: Did the raters watch the ultrasound sequences first before tracking (for both the uninformed and the informed tracking)? From my experiences this is a very helpful way to understand fascicle behaviour before tracking manually during cyclical tasks. Further, did the raters always come back to the initial frame and adjust the fascicle end points in the DIT? This needs to be clarified.

12. Line 230: I would like to see a brief description of the “offset-corrected condition” within the analyses portion of the methods. Please include this.

13. The major concern I have with this analysis (and the accompanied statistical results) is that the authors present absolute lengths only. Indeed (as the authors note in the discussion), the differences in results from DIT for the initial short versus the initial long conditions are an artifact of the absolute fascicle length (and the fact that the fascicle may have been out of the field of view for the initial long condition) - specifically the increased likelihood for differences in identifying total fascicle length between raters. I would like to see further analysis comparing initial short and initial long but using % length change or strain.

RESULTS:

14. Line 244-261: It is my preference to leave out p values if they are not significant.

15. Please include results from analysis of % fascicle length change or strain as noted previously. These could be added to Figure A with a dual Y axis (left side absolute length change and right side % length change from initial or rest).

DISCUSSION:

16. Line 290: I believe this is a typo and should be “this type of error” or “these types of errors”

17. Line 309-318: To my knowledge, the joint movements you are tracking fascicles throughout are not cyclical but rather start at either a plantarflexed position (initial short) and move to dorsiflexed or vice-versa and do not repeat (ie: are not cyclical since they do not come back to initial position). This paragraph begins with discussing cyclical movements- which your results can provide some insight for, but do not assess. Please re-work this paragraph.

18. Line 312: Out of curiosity, how many researchers did you personally communicate with in regards to their preference to start tracking on lengthened fascicles? Were all of these researchers trained in ultrasound by the same group(s)? In my opinion, lengthened typically means that the fascicle end points are out of (or nearly out of) the field of view of the telemed linear transducer, so I prefer shorter lengths.

19. One of my major suggestions for the discussion would be for the authors to include some discussion of comparing these methods to the key frames adjustment that is implemented within Ultratrak. The key frames adjustment would qualify as a type of ‘informed tracking’ and seems to be the gold standard for many research groups tracking fascicles during dynamic tasks or dynamometer contractions. In particular, the first author of this manuscript has used Ultratrak (so presumably the key frames correction) in his previous studies.

20. Line 354-355: As previously mentioned, I would like to see this analysis added to the manuscript. The statement “ this effect may be cancelled when calculating differences relatively to the absolute fascicle length” is, in my opinion, the major limitation to this study which the authors have the data to perform the analysis on.

21. Line 360: I am not sure what you mean by “in a relative sense.” I believe that you mean that normalized length changes or strains are more consistent between raters but absolute length changes differ. Please clarify this sentence in the discussion.

22. Line 365: “both static and active” should be rephrased to either “static and dynamic” or “passive and active”.

23. Line 369-371: The statement “As such, we recommend caution when drawing conclusions from fascicle length comparisons with differences below 10% of absolute fascicle length” seems a bit bold to me. Numerous studies (including some by co-authors of this manuscript) do report fascicle length changes that are less than 10%, for example in the plantarflexors during normal walking. Comparing normalized length changes or strains across studies is, in my opinion a valuable metric.

24. Line 373: “…when comparing absolute fascicle lengths” across what tasks? Please clarify

FIGURES:

25. Figure 1: I would like to see Figure 1 also include the initial short conditions (either on the same plots or as C/D panels. Of interest to readers would be whether the fascicle follows a similar length change trajectory when shortening and lengthening. In addition, as previously mentioned, some methodological details regarding the “initial frame offset” (Fig 1B) is necessary within the manuscript.

OTHER COMMENTS:
26. What was the rationale for passive ankle rotations rather that active (as a % of Fmax or max ankle moment)? From my own ultrasound tracking experiences, I do believe that tracking fascicle length changes in active muscle to be (slightly) more straightforward than passive muscle. If anything, I would lie to see this discussed as a limitation of the current study.

---

## Round 0.2 · Minor Revisions

· Academic Editor

Minor Revisions

Thank you for a detailed and organized response to our reviewer comments. You will find that 2 out of the 3 reviewers are now happy with the quality of the manuscript. One reviewer would like a few, very minor changes. I have decided that this manuscript does not need to go out for further review. I would just like to see reviewer 3's changes addressed and it can be accepted for publication in PeerJ.

Thank you.

Reviewer 1 ·

Basic reporting

I don't have any further comment.

Experimental design

I don't have any further comment.

Validity of the findings

I don't have any further comment.

Comments for the author

I don't have any further comment.

·

Basic reporting

All is good.

Experimental design

All is good.

Validity of the findings

All is good.

Comments for the author

I commend the authors for a nice manuscript.
This has now further improve.

Reviewer 3 ·

Basic reporting

Text:

Line 95-95 The sentence “...because it is a superficial muscle with relatively short muscle fascicles and it has important contributions during tasks such as walking, running and jumping” should include some reference(s).

Line 117: Authors state "From these 28 participants, 5 were randomly selected and single-image ultrasound recordings were taken for an intra-rater reliability test and the UFI analyses"

If I correctly understand the newly added Figure 1, then I believe that following the above copied sentence you need to add one more sentence that says.

“For the DIT comparisons, all 28 subjects were included”

If this is not correct then I am not sure Figure 1 matches the methods text, as from the left side of the figure it appears that all 28 subjects were included in at least the DIT portion of the analysis. If I had not looked at this figure but only read the text, then I would be under the impression that only 5 of the 28 were included for all analysis.

Line 280-281: Please include the specific figure number in Seymore paper in which you are referring to here

Line 321-323: The added sentence “However, other muscle-tendon units often present different fascicle geometries, for example the knee extensors have much longer fascicles and more curvature of the fascicles” requires a reference.


Line 399 Please add “the” to sentence “…by a single rater reliability of the analysis can be increased” should read “…by a single rater the reliability of the analysis can be increased”

Supplemental Files:

What do the data examples AVI videos correspond to? The videos appear to be of a failed auto tracking algorithm? These files need a description bit of text or title to accompany them.

Figure 1 caption: I suggest “schematic” rather than graphical summary (this is not really a graph).

Experimental design

No comment

Validity of the findings

No comment

Comments for the author

No comment

---

## Round 0.3 · accepted · Accept

· Academic Editor

Accept

Thank you for your patience and for addressing our comments so quickly. I look forward to seeing your published article.

Thank you for choosing PeerJ!